# Antibiotic use in mandarin production (*Citrus reticulata* Blanco) in major mandarin-producing areas in Thailand: A survey assessment

Sunicha Chanvatik[1]☯, Siriporn Donnua[2]☯, Angkana Lekagul[1]☯,
Wanwisa Kaewkhankhankhaeng[1]☯, Vuthiphan Vongmongkol[1]‡, Pornpimon Athipunyakom[3]‡,
Saenchai Khamlar[3]‡, Maitree Prommintara[3]‡, Viroj Tangcharoensathien[1]☯*

1 International Health Policy Program, Ministry of Public Health, Nonthaburi, Thailand, 2 Department of Plant Pathology, Faculty of Agriculture at Kamphaeng Saen, Kasetsart University, Bangkok, Thailand, 3 Department of Agriculture, Ministry of Agriculture and Cooperatives, Bangkok, Thailand

☯ These authors contributed equally to this work.
‡ These authors also contributed equally to this work.
* viroj@ihpp.thaigov.net

## Abstract

### Background

Antimicrobial resistance (AMR), one of the major global threats to human security, has serious negative consequences for both health and economies. Excessive and inappropriate uses of antibiotics are the main drivers of the emergence of resistant bacterial strains. In Thailand, antibiotics have been used in citrus production since 2012 to treat citrus greening disease or Huanglongbing disease, despite no antibiotics being registered for use in mandarin. This raises concerns about irrational use of antibiotics, which can cause AMR.

### Objective

To assess the status of greening disease and the use of antibiotics in mandarin production.

### Method

A face-to-face interview survey in 2017 with 221 mandarin growers in two major mandarin-producing areas.

### Findings

Greening disease is one of the most serious diseases in mandarins and farmers in the two major mandarin-producing areas in Thailand used ampicillin, amoxicillin, tetracycline and penicillin to treat it. As no antibiotics are registered for use in plants, farmers used antibiotics (registered with the Thai Food and Drug Administration) for human use, either active pharmaceutical ingredients or finished products. They commonly purchased them from retail pharmacies or agrochemical suppliers. Farmers were influenced to use antibiotics by their orchard neighbours and advice from a few academics. The farmers injected antibiotics into

**Data Availability Statement:** All relevant data are within the paper and its Supporting Information files.

**Funding:** This study was supported by funding from the Food and Agriculture Organization of the United Nations (Grant number: LOA/RAP/2017/17). The funder had no role in the study design, data collection and analysis, decision to publish, or preparation of this manuscript.

**Competing interests:** The authors have declared that no competing interests exist.

the tree trunks approximately three to four times a year and stopped for more than two months before harvesting for in-season fruits.

## Conclusion

Antibiotics registered for human use are being applied to control greening diseases. We recommend scaling up sustainable disease control measures and curtail the use of antibiotics through close and effective dialogue among 'One Health' partners.

## Introduction

Antimicrobial resistance (AMR) is a global problem and recognized as one of the top global health security threats as it spreads through food, water, animals and people, resulting in public health crises, environmental pollution and economic loss [1–4]. Inappropriate use of antimicrobials in agriculture facilitates the emergence of resistant bacterial strains [5]. In response to the Global Action Plan on AMR, countries should monitor the use of antibiotics and situations of antibiotic resistance through effective surveillance systems and informed policy interventions [1].

Antibiotics are used to control bacterial diseases in humans, animals and plants [6]. Although there is unclear evidence on the link between AMR in humans and antimicrobial use in plants [7, 8], the 2018 report on AMR raises serious public concern about antibiotics sprayed on crops, which affects the environment and potentially increases the risk of AMR [9]. However, spraying has been used for citrus greening disease or Huanglongbing disease in the United States since 2016 [10].

Citrus greening disease was first reported in Southern China in 1919 and has spread to different countries in Africa, Oceania, America and Asia including Thailand [11–15]. Each year 10–15% of mandarin trees are destroyed by citrus greening disease in the northern region of Thailand [13, 14, 16, 17]. Greening disease is transmitted by an insect vector, the Asian citrus psyllid (*Diaphorina citri*) [18]. The symptoms of infected plants are yellow shoots, leaf mottling, poor productivity and finally death of the entire plant. Despite no definitive treatment [15], a few control measures are used including psyllid control, elimination of infected plants and antibiotics. A recent study reported that antibiotics such as penicillin, streptomycin and oxytetracycline hydrochloride are more effective in treating greening disease than eight plant defense activators [19]. Literature confirms that ampicillin and penicillin are effective treatment of greening disease [20].

In Thailand, the use of antibiotics in citrus was first piloted in 1998 and then widely practiced for managing greening disease through trunk injection since 2012. For trunk injection, farmers drill hole(s) on the trunk 50–70 cm above the ground, and push antibiotics solution by syringes or drip in plastic bottles into the tree through the hole(s). Trunk injection of a mixture of streptomycin (250 mg/l), ampicillin (2.5 g/l), penicillin G (2 g/l), and Bacicure® (2 g/l) provided the highest efficiency in reducing and suppressing the Las-bacterium population in the field experiments [21]. Injection of 15 syringes per tree of 500 mg ampicillin/20ml water/syringe on pomelo, Koaw-Tang-Kwao cultivar every 2 months reveal high recovery from greening symptom from 4 through 10 months after injection [22].

In addition, citrus propagule dipping and foliar spraying methods were also sporadically practiced [23]. There is no legal approval for antibiotic use in plants from the Department of Agriculture (DOA) and the previous license that existed for using other chemicals has expired.

In 2016, Thailand adopted and implemented the national strategic plan on AMR, which was endorsed by the Cabinet. In parallel, monitoring antibiotic consumption and baseline indicators were developed. The national plan strengthens antimicrobial stewardship and prudent use of antibiotics in all sectors [24].

In response to prudent use in agriculture, this study assesses the status of greening disease, its magnitude and pattern and the sources of antibiotics use in mandarin (*Citrus reticulata* Blanco) production. A better understanding of antibiotic use provides evidence to inform policy to address the disease and to combat AMR.

## Materials and methods

### Study sites and sample population

In September and October 2017, we conducted interview surveys in two major mandarin-producing provinces in Thailand: Chiang Mai and Pathum Thani. According to the 2016 database held by the Department of Agriculture Extension (DOAE), both provinces were the largest cultivated area in the north and central regions. We further selected the highest production district from each of the provinces as study sites (district A in Chiang Mai and district B in Pathum Thani).

Based on our preliminary field visit to Chiang Mai and the interviews with five farmers and agricultural officers from Chiang Mai Agricultural Research and Development Centre, DOA, in August 2016, we estimated 80% of mandarin orchards used antibiotics. Sample size calculations (OpenEpi, version 3) were based on the DOAE-registered orchard database in 2016, with 95% confidence level and an expected error of 5%. There were 1,445 orchards in district A of Chiang Mai and 177 orchards in district B of Pathum Thani. A total of 221 orchards were selected as samples for this study.

The sampling frame, obtained from the DOAE updated orchards list as of January 2017, was used as a basis for systematic random sampling but failed as the list was not updated. It gave either wrong addresses, or farmers had moved out from the area, or growing other plants. A new list of farmers was revised on-site by local DOA staffs in both districts. These farmers, included both active and non-active farmers (who closed their mandarin orchards due to greening disease), were purposively selected and invited for a face-to-face interviews. We invited them until the 221 samples in both district had completed the interview.

### Sampling and data collection

All farmers in the revised list were contacted to participate in the study by local agriculture officers. The interview survey took place in a neutral meeting room of the sub-district administrative organization. For those who could not join, the interview survey took place at the farmers' houses. All respondents signed the consent form prior to the interview. Face-to-face interviews were conducted in Thai by the trained research team, and took 30 to 40 minutes to complete. Interviews in both districts were conducted by the same research team. All data were anonymous with strict confidentiality; the questionnaire forms had neither names nor identifiers. The study was approved by the Institute for the Development of Human Research Protections (IHRP) for research ethics clearance (Ref.no.IHRP2017026).

Dose of antibiotic treatment is measured by part per million (PPM); where 1 PPM equal to 1 mg of antibiotics per litre of water. Farmers were asked a) if they used active ingredient, how many grams of powder were mixed with how many litres of waters; or b) if they used finished products such as capsules, what are the milligram of antibiotics per capsule, how many capsule (s) were added with how many litres of water.

### Questionnaire survey

The survey questionnaire, developed in the Thai language, was peer reviewed by two external experts from different fields (pharmacologist and plant expert) for content validity, the logic and clarity of the content. It was pilot tested with 20 farmers, which were randomly selected from orchard growers in two provinces, before the final version was reached with an aim to improve the content validity. We found the contents were easily understood and accurate with no controversy. There was minor amendment of the questionnaire after piloting. The face-to-face interview questionnaire comprised six sections 1) general information; 2) mandarin production process; 3) status of greening disease and management response; 4) use of antibiotics; 5) production cost; 6) production volume and market value. All questions covered a recall of information over the past 12 months. This paper focused on the size of orchards affected by greening disease, disease management and use of antibiotics.

### Data analysis

Doses of antibiotics were reported in parts per million (PPM) unit, which are calculated from one milligram of antibiotic powder per one litre of water (1 PPM = 1mg/L). Survey data were entered onto an Excel spreadsheet. Data were analyzed by SPSS software, version 20 (SPSS Inc., USA) using descriptive statistics including medians and inter-quartile ranges (IQR).

## Result

### Characteristics of surveyed mandarin orchards and farmers

The survey was conducted with farmers from 221 mandarin orchards; 143 in district A of Chiang Mai and 78 in district B of Pathum Thani. All invited farmers participated in this study with zero refusal. Half of respondents' orchards are of medium size (10 to 49 Rai, equal to 1.6 to 7.8 hectares) (54.3%). Three quarters of respondents were not members of mandarin grower co-operatives (74.2%). Half of the farmers were primary school educated (49.8%). The farmer respondents had an average of 16 years experience in mandarin production. Of the total 221 respondents, 87.3% (193) were still active in mandarin production; while the remaining were not active farmers; they had abandoned growing mandarin because of greening disease. See Table 1.

### Status of greening disease in 2016

Greening disease was the most commonly reported challenge in both districts, as 87.6% of total active orchards (169/193) were affected, followed by canker disease (81.9%), scab (59.1%) and root rot (53.4%). There were also other fungal diseases with citrus gummosis (18.7%), tristeza (11.9%), and mandarin pests also reported (Fig 1).

Researchers explored the size of the problem among those total farmers who reported their orchards were affected by greening disease (169). Greening disease affected less than 30% of total plants in 69.2% of sample orchards, between 30% and 50% in 16.6% of sample orchards, and more than 50% to 100% in 14.2% of sample orchards.

### Management of greening disease

Of the 169 orchards affected by greening disease, six methods were used, with multiple methods are commonly applied: 95.3% of farmers used antibiotics; 65.1% cut and burned the affected branches; 20.1% cut the affected branches without burning; 27.8% partially eradicated the severely affected sections; 33.1% used chemical control; and only 1.8% applied total eradication, (Fig 2).

**Table 1. Number and profiles of respondents in questionnaire survey covered by the study.**

| Demographic data | Total respondents |
|---|---|
| **Total** | **221** |
| Age (Mean, range; year) | 51 (21–78) |
| Gender | |
| - Male (%) | 157 (71.0%) |
| - Female (%) | 64 (29.0%) |
| Education level | |
| - Uneducated (%) | 4 (1.8%) |
| - Primary school (%) | 110 (49.8%) |
| - Secondary school (%) | 66 (29.8%) |
| - Vocational Certificated (%) | 11 (5.0%) |
| - Undergraduate (%) | 28 (12.7%) |
| - Postgraduate (%) | 2 (0.9%) |
| Status | |
| - Owner (%) | 199 (90.1%) |
| - Cousin (%) | 10 (4.5%) |
| - Employee (%) | 12 (5.4%) |
| Work experience in the orchard (Mean, range; year) | 15.6 (0–59) |
| Active farmers in 2016 | |
| - No. of active farmers in 2016 | 193 (87.3%) |
| - No. of non-active farmers in 2016 | 28 (12.7%) |
| Membership in Co-operative relevant to mandarin | |
| - Membership | 57 (25.8%) |
| - Non-membership | 164 (74.2%) |
| Cultivated area (Rai*) | |
| - Small (< 10 Rai) | 76 (34.4%) |
| - Medium (10–49 Rai) | 120 (54.3%) |
| - Large (> 49 Rai) | 25 (11.3%) |

Note

*1 hectare equivalent to 6.25 rai.

**Use of antibiotic: Type, volume, dosage and form.** In 2016, of 193 active orchards in the survey, 184 (95.3%) reported using at least one antibiotic. Only nine out of 193 (4.7%) reported they did not use antibiotics.

From this survey, only four antibiotics were used: ampicillin, amoxicillin, penicillin and tetracycline which belong to Beta-lactams and Tetracycline classes. Of these 4 antibiotics, ampicillin was most commonly used in 177 out of 184 orchards, or 96.2% of total reported use; see Table 2.

Three antibiotics belonging to beta-lactams class (ampicillin, amoxicillin and penicillin) are classified by the 2017 WHO as high priority, critically important antimicrobials (CIA) for human use, while antibiotics in the tetracycline class are classified as highly important antimicrobials [25]. Of 184 orchard farmers who reported using antibiotics, 13 orchards used a combination of ampicillin and tetracycline, and two orchards used a combination of ampicillin and amoxicillin.

Farmers added water into ampicillin powder and injected it into the trunk. The concentration ranged from 100 to 333,333 part per million (PPM), with a median concentration of 15,000 PPM (IQR 2,500 PPM). One PPM equals one milligram of antibiotics per one liter of water.

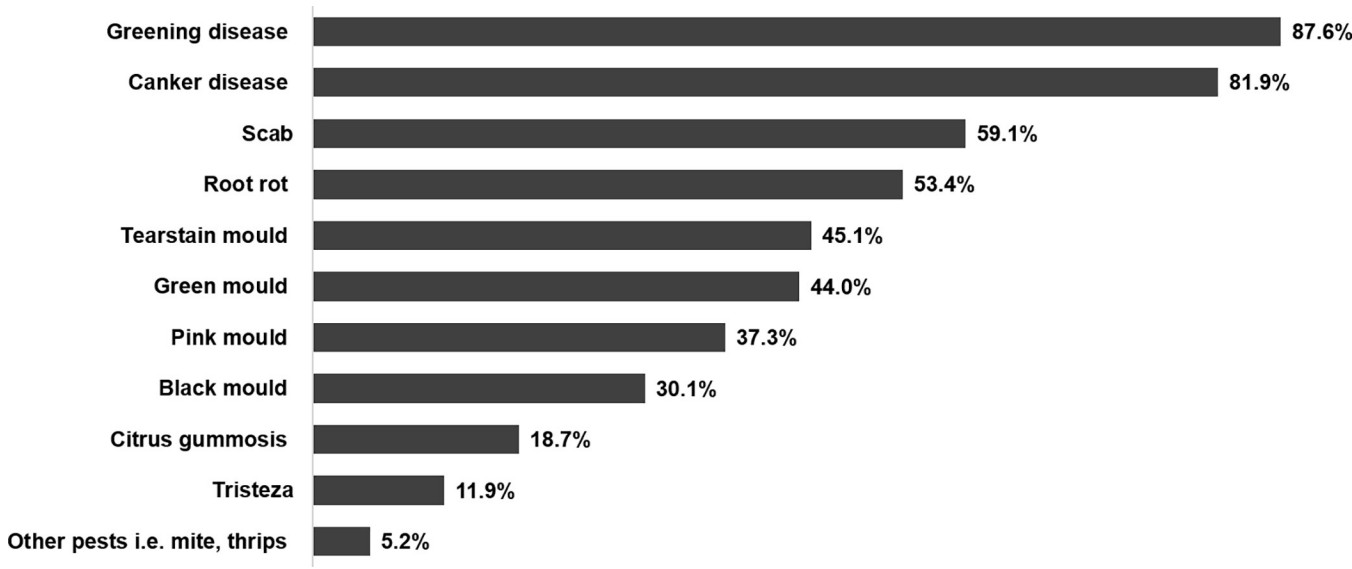

**Fig 1. Common diseases in surveyed mandarin orchards in Thailand, 2016 (Multiple answers).**

For the amoxicillin injection, the median dose was 7,500 PPM, with ranges between 7,500 and 100,000 PPM (IQR = 92,500). The dose for tetracycline was much lower than ampicillin; its median was 500 PPM, with ranges between 100 and 4,166 PPM (IQR = 750). The median dose of penicillin was 12,500 PPM, with ranges between 12,500 and 150,000 PPM (IQR = 72,500).

Farmers reported that the median volume of water-soluble antibiotics they injected was around 40 ml for a small tree, with ranges between 5 and 400 ml (SD = 85.4), while a larger tree received around 400 ml, with ranges between 20 and 2,000 ml (SD = 261).

Factors affecting the volume of the antibiotics injection were the size of the tree (85.3%) and the age (34.8%), and over a quarter of respondents (27.7%) said the volume depended on the symptoms of greening disease. There were a small number of respondents, less than 1%, who reported that the number of affected branches determined the volume of injection.

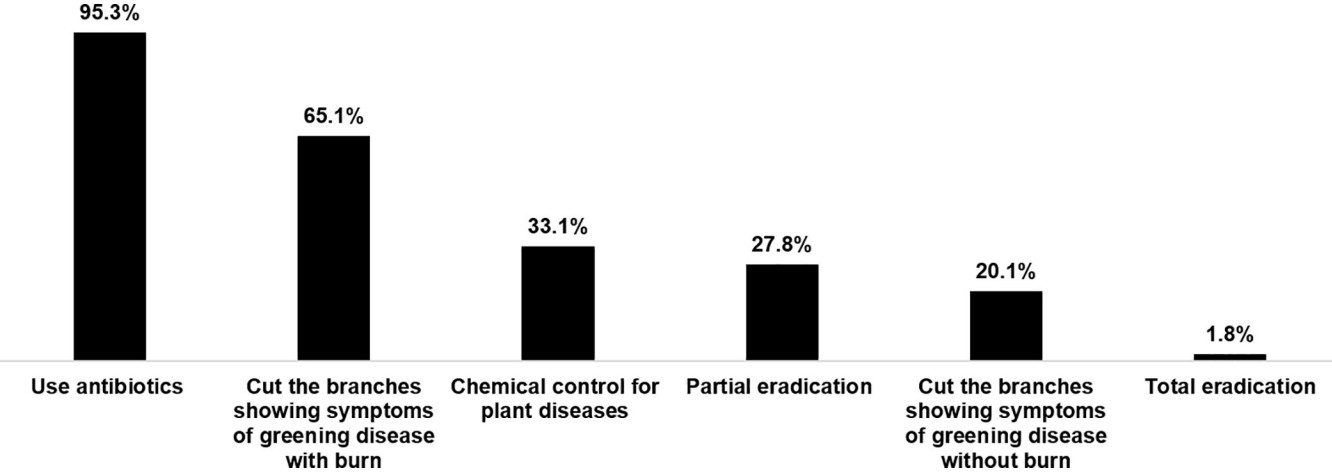

**Fig 2. Management of greening disease in surveyed mandarin orchards in Thailand, 2016 (Multiple answers).**

**Table 2. Antibiotics used in mandarin orchards for treatment of greening disease in 2016.**

| Class | Antibiotics | Number of orchards in which antibiotic is used (n = 184) * | On the WHO's critically important antimicrobials list (2017) | Dose; Median PPM (min-max), IQR |
|---|---|---|---|---|
| Beta-lactams | ampicillin | 177 | High priority critically important antimicrobials | 15,000 (100–333,333), 2,500 |
| | amoxicillin | 3 | High priority critically important antimicrobials | 7,500 (7,500–100,000), 92,500 |
| | penicillin | 5 | High priority critically important antimicrobials | 12,500 (12,500–150,000), 72,500 |
| Tetracyclines | tetracycline | 14 | Highly important antimicrobials | 500 (100–4,166.67), 750 |

Note

*some orchards used more than one antibiotic

More than half (57.1%) of respondents who used antibiotics, had sourced antibiotics in the form of a finished product, in particular capsules bought from pharmacies. However, nearly a half (42.9%) of respondents used antibiotics in the form of active pharmaceutical ingredients (API). Ampicillin was the only antibiotic for which respondents used both forms of finished product and API.

**Application methods.** Several application methods included injection, with foliar spraying and soil drenching at tree trunk being less common. Of a total of 184 orchards which reported use of antibiotics, 168 (91.3%) used only one method, and 16 (8.7%) used more than one method.

Direct trunk injection was the most commonly used method (99.5%), followed by foliar spraying (5.4%) and soil drenching at tree trunk (4.3%). For injections, more than two thirds (69.9%) of respondents used a dripping technique, and 30.1% used a syringe and pumping technique.

The site of injection also varied; most respondents injected antibiotics into the trunk (89.1%), followed by disease-affected main branches (54.3%), small affected branches (22.3%) and other areas such as main branches with no symptoms of greening disease (1.6%).

The first application of antibiotics to a tree varied; the most common time was when the tree was aged 1–2 years (45.4%), then between 2–3 years old (19.1%), less than one year old (18.6%) and more than three years old (16.9%). In terms of frequency, farmers applied the treatment every 3–4 months (69.9%), followed by 5–6 months (21.9%), sometimes as frequently as 1–2 months (7.7%) and less frequently at 12 months interval (0.5%).

In terms of the withdrawal period prior to harvesting, 56.8% of respondents stopped antibiotics application for more than two months, 24.1% stopped at more than one month, and 8.7% less than one month. However, 10.4% of farmers could not give an answer on antibiotics withdrawal, as it depended on the cycle of injection.

**Source of antibiotics.** More than a half of active orchards (60.1%) bought antibiotics from private pharmacies; while almost a quarter (23.5%) bought them from agrochemical suppliers. A minority bought antibiotics from pharmacy retailers (7.1%), neighbouring orchards (6%), other farmers outside the district (2.7%) and the academic sector (0.6%).

Sources of antibiotics by pharmaceutical forms also differ. For those who used finished product, a vast majority of farmers sought them from pharmacies (80%), followed by agrochemical suppliers (15.2%), neighbouring orchards (3.8%) and pharmacy retailers (1%). For those who used antibiotics in the form of API, farmers reported that they bought API from agrochemical suppliers (34.6%) and pharmacies (33.3%), followed by pharmacy retailers

**Fig 3. Source of antibiotic drugs in mandarin orchards in Thailand, 2016.**

(15.4%), neighbouring orchards (9%), farmers in other districts (6.4%) and academic sector (1.3%). See Fig 3.

**Information about use of antibiotics for greening disease.** Farmers who used antibiotics received information about antibiotics for controlling greening disease from other farmer peers (65.6%), advice from peers in different districts (10.4%), the academic sector (12.6%), private pharmacies (4.9%), agrochemical suppliers (2.7%), pharmacy retailers (2.2%) and other sources such as advertising on the radio (1.6%) (Fig 4).

## Discussion

Several publications reported the use of antibiotics in crops [1, 25], though only a few provide evidence on antibiotic use in mandarin production [4]. To our knowledge, this study is the first from Thailand which assesses at great detail the use of antibiotics in mandarin production in the context of AMR and the national strategic plan requirement to optimize use in human and agriculture sectors.

### Management of greening diseases

In the past years, the government adviced farmers to control greening disease by partial eradication of infected branches or trees and chemical control of vector. The government also supported farmers with disease-free propagules for replacement. However, these control efforts

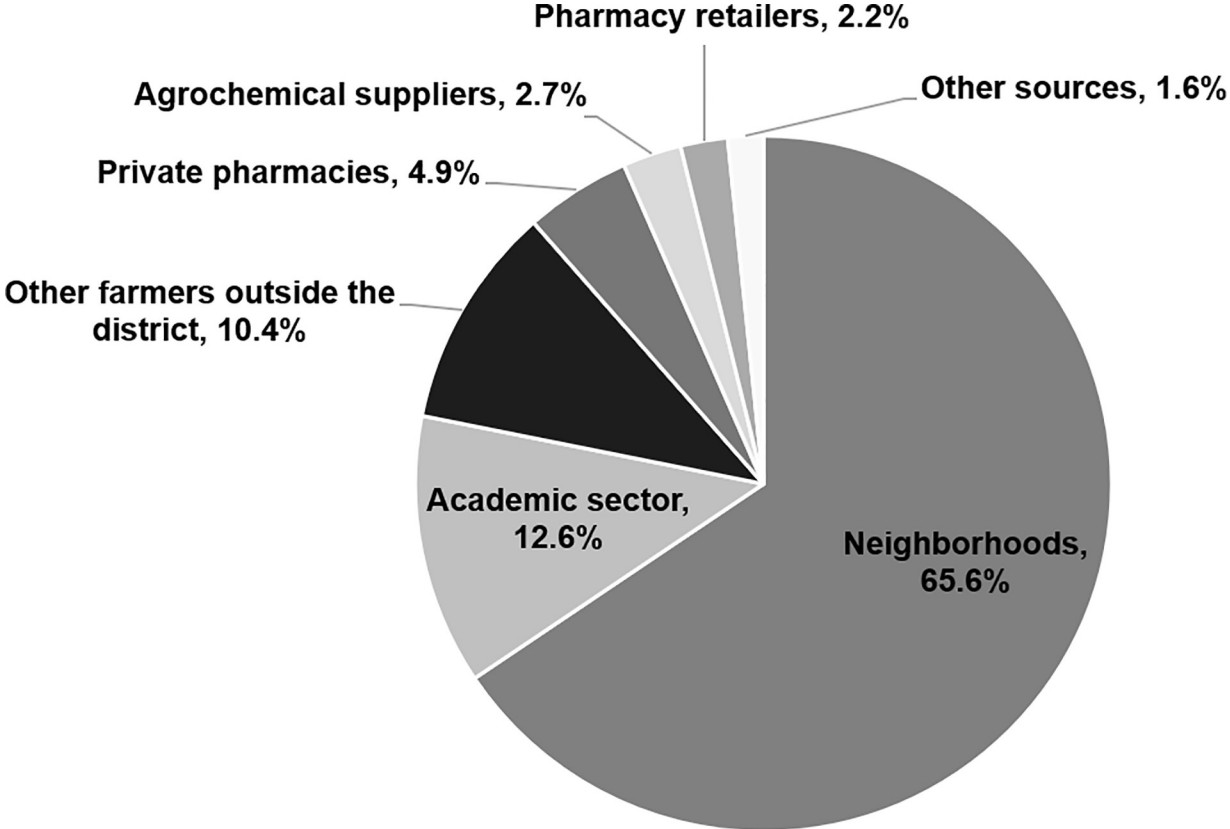

**Fig 4. Information about using antibiotics in mandarin orchards in Thailand, 2016.**

were not successful; as there was no collective and synchronized effort by farmers to control the vectors and removal of affected plants in the whole affected areas; that the vectors move from the non-controlled areas to infect the controlled areas. The abandoned farms with heavily infected trees are the major pool of disease propagation through vectors.

Control efforts in other countries are similar to that applied in Thailand but more intensive and collective. These are vector control, removal of infected trees, providing enhanced nutrition by foliar sprays of readily absorbable nutrients and phytohormones, regulating soil pH to enhance nutrient uptake and precision irrigation using soil moisture sensing to the needs of HLB-affected trees [26–29].

### Use, misuse of antibiotics and its negative consequences

Only four types of antibiotics were reported in the studied areas. The study highlights that mandarin farmers modify and adapt the use of antibiotics such as concentration, volume, frequency, route of administration and combination of antibiotics. In addition, information from other farmers, academia, pharmacists at retail pharmacies and sellers at agrochemical suppliers also influenced farmers' decisions. Despite different advice, farmers adjusted the treatment protocol according to the results of their experiments.

A wide range of concentration has been reported by farmers. As there is no antibiotic registered for use in mandarin, there are no guidelines from the Ministry of Agriculture and Cooperatives on concentration, volume, frequency and withdrawal period. In the first few years between 1998 and 2012, farmers followed strictly the recommended frequency and dosage

when tetracycline was piloted by certain academia. However, tetracycline is phytotoxic, causing more damage than the disease itself [30]. When tetracycline was replaced by non-phytotoxic ampicillin, farmers learned the application techniques through experience.

Some farmers further adapted the new application of antibiotics such as foliar spray and soil drenching at the tree trunk similar to the use of pesticides, insecticides and chemical agents. All these adverse methods could potentially contaminate antibiotics in the environment through soil, air and in the watershed. This can have a serious negative impact on the health of humans, animals and the environment.

Dosages and frequency are also determined by various factors such as the magnitude of disease, and the prevalence and control of psyllid in the areas. Since there is no technical advice from the agriculture authority, responses to greening disease are not systematic and effective. Antibiotic treatment is ineffective when the trees become five to eight years old or when there is extensive infestation. At this stage, the farmers have no choice except total eradication [12].

### Easy access to antibiotics and weak regulation of API

The Thailand Drug Act BE 2510 (1967) authorizes API to be sold to manufacturers to produce finished products. API cannot be sold or used directly in humans, animals or plants due to its high concentration and potential for abuse. Respondents reported the use of antibiotics in the form of API purchased from agrochemical suppliers and retail pharmacies. During visits to the orchards, researchers eye-witnessed cartons of 25 kilograms ampicillin in the form of API in several orchards.

Farmers can purchase large quantities of the four finished forms of antibiotics from retail pharmacies without prescription. Antibiotics are classified by the Thai FDA as a "dangerous drug" which does not require prescription, although antibiotics do need to be dispensed by licensed pharmacists. This results in farmers have easy access to antibiotics, which can be inappropriately used in the management of greenings disease; in particular ampicillin, amoxicillin and penicillin which are classified as "high priority CIA" and tetracycline as "highly important antimicrobials". The government is concerned about the potential excessive or sub-optimal use that can trigger resistance of bacterial pathogens in the environment as a result of exposure to these antibiotics.

### Limitation of study

The sampling method is the main limitation of this study. Based on the DOAE's database, there were orchard growers in 14 districts of Chiang Mai and 3 in Pathum Thani. Due to a limited survey budget, we decided to select the highest production district from each province as study sites. The two purposively selected districts do not represent the entire province on disease management and use of antibiotics. The two sites are highly infected by greening disease and are also sites with high users of antibiotics. Interpretation is made with care and this study cannot be generalized to a national scale reflecting antibiotic use in mandarin production.

There is one outlier of very high concentration of antibiotics, up to 333,333 PPM (see Table 2). This concentration is correct after re-confirmation with the respondent. The preparation is 1,000 gram of ampicillin in active pharmaceutical ingredients form in 3 Litres of water; which is equivalent to 333,333 PPM. It is unclear if the reported 1000 gram by the farmer is accurate.

### Potential sustainable solutions

Although tree pruning is effective to control fungal diseases, it is not effective for the prevention of greening disease [31]. Eradication of infested trees can reduce the source of inoculum,

but it is costly to farmers and requires cooperation by all farmers in the affected area; the added benefit being that the government may consider compensation. Full eradication can also prevent infection to Rutacious species, as *Candidatus* Liberibacter asiaticus can infect lime (*Citrus aurantifolia*), pomelo (*Citrus maxima*) and jasmine orange (*Murraya paniculata*) [32]. Eradication is a successful and sustainable solution to citrus canker in Florida [33]. Replanting the disease-free plants are needed after eradication. Net screen house nursery to prevent vector is necessary for disease free plant production. Parallel interventions are required to control vectors. Partial eradication of infected branches or trees and re-planting disease free propagules, though feasible and less costly, it is not comprehensive and sustainable.

## Recommendations

Given the evidence generated from this study, we propose three key recommendations.

Firstly, control of API distribution is needed through effective tracking mechanisms to prevent leakage into agriculture sector. Warning of legal actions for law violations should be issued to retail pharmacies, agrochemical suppliers (which are not licensed to sell any type of medicines) and pharmaceutical retailers selling API directly to consumers. Mandarin farmers should be informed that inappropriate use of antibiotics in mandarin can induce AMR in the pathogens in environment through contamination of antibiotics in water and soil; while at the same time Ministry of Agriculture needs to establish sustainable solution to greening diseases and replace use of antibiotics.

Secondly, addressing the complex issue such as greening disease requires multi-sectoral, multi-disciplinary collaboration through One Health approach. This includes a) Department of Agriculture, Ministry of Agriculture and Cooperatives responsible for controlling plant disease such as vector control, promotion of low cost disease free propagules replacing mandarin in the affected areas, providing technical support and advices to farmers; b) Food and Drug Administration regulates distribution of antibiotics; and local health centers create awareness of AMR and knowledge about consequences of inappropriate use of antibiotics; c) Department of Environmental Quality Promotion responsible for environment protection from antibiotic contamination in the water and soil can provide advice on safe disposal of antibiotics; and d) International Health Policy Program, of the Ministry of Public Health responsible for health systems and policy research can convene policy dialogues among three sectors: agriculture, health and environment to find best possible solutions. The One Health approach benefits mandarin farmers for example, control of disease and improve productivity without excessive use of antibiotics; increased awareness on AMR and proper use of antibiotics; and protect their environment from the emergence of resistant pathogens. Further, policy decisions should be informed by evidence on whether antibitoics should be registered for use in plants and to ensure better control of antibiotic use, dosing, volume, interval and withdrawal period. Decisions should take into account the pros and cons of antibiotic use in the context of AMR and selective pressure against the emergence of AMR pathogens in the environment.

Thirdly, there is a need to rapidly institute sustainable solutions against greenings disease. This requires combining three synergistic measures: scaling up production capacities of disease-free propagules for replacement; effective psyllid control such as physical (a net with appropriate mesh size for a closed system) or chemical (petroleum oil and insecticides) measures; and eradication of affected orchards to minimize the sources of inoculum. A closed system requires high capital investment and may not be feasible in certain geographically challenging areas such as hilly terrains. All these combined efforts must be implemented across whole plantation areas as vectors can carry *Candidatus* Liberibacter asiaticus from the infected to non-infected plants. Total eradication may require government subsidies, in particular

when farmers are small orchard holders. The Department of Agriculture is responsible for certifying and re-certifying nurseries to produce disease-free propagules through training and supervision. Academia plays a pivotal role to support this effort.

## Supporting information

**S1 File. Certificate of approval from ethics committee.**
(PDF)

**S2 File. Questionnaire English version.**
(PDF)

**S3 File. Questionnaire Thai version.**
(PDF)

**S4 File. Raw data for analysis.**
(XLSX)

## Acknowledgments

We are grateful to agricultural officers from Chiang Mai and Pathum Thani Agricultural Research and Development Centre, Department of Agriculture for their contribution to this work.

## Author Contributions

**Conceptualization:** Sunicha Chanvatik, Siriporn Donnua, Angkana Lekagul, Wanwisa Kaewkhankhaeng, Pornpimon Athipunyakom, Saenchai Khamlar, Viroj Tangcharoensathien.

**Data curation:** Sunicha Chanvatik, Wanwisa Kaewkhankhaeng, Saenchai Khamlar.

**Formal analysis:** Sunicha Chanvatik, Wanwisa Kaewkhankhaeng, Vuthiphan Vongmongkol.

**Funding acquisition:** Viroj Tangcharoensathien.

**Investigation:** Sunicha Chanvatik, Pornpimon Athipunyakom.

**Methodology:** Sunicha Chanvatik, Siriporn Donnua, Angkana Lekagul, Pornpimon Athipunyakom, Saenchai Khamlar, Viroj Tangcharoensathien.

**Project administration:** Wanwisa Kaewkhankhaeng, Saenchai Khamlar.

**Resources:** Siriporn Donnua, Pornpimon Athipunyakom, Saenchai Khamlar, Maitree Prommintara, Viroj Tangcharoensathien.

**Software:** Wanwisa Kaewkhankhaeng, Vuthiphan Vongmongkol.

**Supervision:** Siriporn Donnua, Vuthiphan Vongmongkol, Pornpimon Athipunyakom, Maitree Prommintara, Viroj Tangcharoensathien.

**Validation:** Sunicha Chanvatik, Siriporn Donnua, Angkana Lekagul, Viroj Tangcharoensathien.

**Visualization:** Wanwisa Kaewkhankhaeng.

**Writing – original draft:** Sunicha Chanvatik.

**Writing – review & editing:** Siriporn Donnua, Angkana Lekagul, Maitree Prommintara, Viroj Tangcharoensathien.

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
