## [Decision Letter · Decision Letter 0]

5 Sep 2019

PONE-D-19-19263

Antibiotic use in mandarin production (Citrus reticulata Blanco) in major mandarin-producing areas in Thailand: a survey assessment

PLOS ONE

Dear Dr. Tangcharoensathien,

Thank you for submitting your manuscript to PLOS ONE. After careful consideration, we feel that it has merit but does not fully meet PLOS ONE’s publication criteria as it currently stands. Therefore, we invite you to submit a revised version of the manuscript that addresses the points raised during the review process.

We would appreciate receiving your revised manuscript by Oct 20 2019 11:59PM. To enhance the reproducibility of your results, we recommend that if applicable you deposit your laboratory protocols in protocols.io, where a protocol can be assigned its own identifier (DOI) such that it can be cited independently in the future. For instructions see: http://journals.plos.org/plosone/s/submission-guidelines#loc-laboratory-protocols

We look forward to receiving your revised manuscript.

Kind regards,

Richard Mankin, Ph. D.

Academic Editor

PLOS ONE

Journal Requirements:

2. Please include copies of the survey questions or questionnaires used in the study, in both the original language and English, as Supporting Information, or include a citation if they have been published previously.

3. Please ensure that you refer to Figures 1, 2 and 4 in your text as, if accepted, production will need this reference to link the reader to the figures..

Additional Editor Comments (if provided):

Please address the following comments from reviewers and editor:

Comments from reviewer 1

The manuscript by Chanvatik et al., described the current status of citrus HLB and the utilization of antibiotics to control this disease in two Thailand provinces using questionnaire method. The authors used the appropriate analysis methods. This is a very interesting work. Based on the analysis results, we can know that the citrus HLB and use of antibiotics to control this disease are very common in Thailand, which will lead the AMR concern. The authors pointed out the limitation of this study and discussed the potential sustainable solutions and recommendation for the disease control of citrus HLB in Thailand. I have several minor issues as following.

2. The data listed in table 1 is mean or median value? Please clarify.

3. Reference 1 and 6 were repeated.

4. I suggested the authors provide the raw data for each analysis as supplemental data.

Comments from reviewer 2

This manuscript reports on the use of a survey of mandarin farmers in Thailand to assess their practices in controlling citrus greening disease. What was found was that antibiotic use was highly prevalent, although no antibiotics were registered for use in disease control. Apparently, information from neighboring farmers was the main driver in influencing individual farmers to use antibiotics. The survey did uncover sources of antibiotics to the growers, including their ability to obtain active pharmaceutical ingredients (API). One point that was not clear was that the authors listed doses of antibiotics used but I am not sure they are correctly listing these. For example, what is the concentration of ampicillin in API or in pills that are used by the farmers?

While the results are clearly disturbing, I am not sure if the results of a single survey warrant publication as a full manuscript. Also, it is not clear that the authors are that well versed in agricultural techniques etc. I did not see any statistical analysis of the results. Finally, I am not sure if all of the necessary requirements are present for conducting a survey and utilizing survey information in a publication.

Ln 50 – probably a brief description and some references for trunk injection are needed as some readers of the manuscript would likely not be familiar with this practice. Also, is ther anything known about the relationship between injected chemicals and residues in fruit?

Lns 96-97 – need more information on the peer review conducted on the survey questionnaire

Table 2 – some of these concentrations seem outrageously high (333,333 ppm?). Do the authors have information on how reliable these estimates are from individual growers? Also, are the antibiotics even soluble at these concentrations? This information should be included.

Ln 187 – again, references detailing these techniques should be included.

Ln 293 – why is there no recommendation about informing and educating farmers that these practices are illegal and can potentially pollute the environment and influence the occurrence of transferrable antibiotic resistance

Comments from editor.

One Health platform needs to be defined somewhere in the text along with a listing of benefits it can provide to orchard owners.

The audience appears to be policy makers who might read the paper and be led to take action to restrict usage of antimicrobials against HLB or establish greater dialog with stakeholders. Possibly it includes persons involved in extension, development of control methods, or wide-area management, but there is little discussion of past efforts in these areas. Is that because previous efforts have been limited? In that case, you might discuss efforts that have occurred in other countries.

At line 225, Reference 26 may actually be 25.

Reviewers' comments:

Reviewer's Responses to Questions

**Comments to the Author**

1. Is the manuscript technically sound, and do the data support the conclusions?

Reviewer #1: Yes

Reviewer #2: Partly

2. Has the statistical analysis been performed appropriately and rigorously? 

Reviewer #1: N/A

Reviewer #2: No

3. Have the authors made all data underlying the findings in their manuscript fully available?

Reviewer #1: No

Reviewer #2: No

4. Is the manuscript presented in an intelligible fashion and written in standard English?

Reviewer #1: No

Reviewer #2: Yes

5. Review Comments to the Author

Reviewer #1: The manuscript by Chanvatik et al., described the current status of citrus HLB and the utilization of antibiotics to control this disease in two Thailand provinces using questionnaire method. The authors used the appropriate analysis methods. This is a very interesting work. Based on the analysis results, we can know that the citrus HLB and use of antibiotics to control this disease are very common in Thailand, which will lead the AMR concern. The authors pointed out the limitation of this study and discussed the potential sustainable solutions and recommendation for the disease control of citrus HLB in Thailand. I have several minor issues as following.

1. Please improve the grammar for the current manuscript.

2. The data listed in table 1 is mean or median value? Please clarify.

3. Reference 1 and 6 were repeated.

4. I suggested the authors provide the raw data for each analysis as supplemental data.

Reviewer #2: This manuscript reports on the use of a survey of mandarin farmers in Thailand to assess their practices in controlling citrus greening disease. What was found was that antibiotic use was highly prevalent, although no antibiotics were registered for use in disease control. Apparently, information from neighboring farmers was the main driver in influencing individual farmers to use antibiotics. The survey did uncover sources of antibiotics to the growers, including their ability to obtain active pharmaceutical ingredients (API). One point that was not clear was that the authors listed doses of antibiotics used but I am not sure they are correctly listing these. For example, what is the concentration of ampicillin in API or in pills that are used by the farmers?

While the results are clearly disturbing, I am not sure if the results of a single survey warrant publication as a full manuscript. Also, it is not clear that the authors are that well versed in agricultural techniques etc. I did not see any statistical analysis of the results. Finally, I am not sure if all of the necessary requirements are present for conducting a survey and utilizing survey information in a publication.

Ln 50 – probably a brief description and some references for trunk injection are needed as some readers of the manuscript would likely not be familiar with this practice. Also, is ther anything known about the relationship between injected chemicals and residues in fruit?

Lns 96-97 – need more information on the peer review conducted on the survey questionnaire

Table 2 – some of these concentrations seem outrageously high (333,333 ppm?). Do the authors have information on how reliable these estimates are from individual growers? Also, are the antibiotics even soluble at these concentrations? This information should be included.

Ln 187 – again, references detailing these techniques should be included.

Ln 293 – why is there no recommendation about informing and educating farmers that these practices are illegal and can potentially pollute the environment and influence the occurrence of transferrable antibiotic resistance?

6. PLOS authors have the option to publish the peer review history of their article (what does this mean?). If published, this will include your full peer review and any attached files.

Reviewer #1: No

Reviewer #2: No

---

## [Author Response · Author response to Decision Letter 0]

18 Oct 2019

Dear Editor, PLOS ONE

Many thanks for the feedbacks by two independent reviewers; we found them very useful which strengthens the manuscript significantly. We had fully reviewed and responded to all comments and suggestions; both in track changes and in clean text versions. This file provides detail point by point responses to their comments. 

We submit the following: 

1. A revised version: clean text version 18 October 2019

2. A track change version showing edits in responses to the reviewers, version 18 October 2019 

3. A response to reviewers’ point by point comments (which is this file) 

4. Four supporting files: 

• S1_file: Ethical approval (word file)

• S2_file: Questionnaire English version (word file)

• S3_file: Questionnaire Thai version (word file) 

• S4_file: Raw data for analysis (Excel file) 

We look forwards to hearing from you if there is anything we need to take further action.

---

## [Editor Report · Decision Letter 1]

31 Oct 2019

Antibiotic use in mandarin production (Citrus reticulata Blanco) in major mandarin-producing areas in Thailand: a survey assessment

PONE-D-19-19263R1

Dear Dr. Tangcharoensathien,

We are pleased to inform you that your manuscript has been judged scientifically suitable for publication and will be formally accepted for publication once it complies with all outstanding technical requirements.

With kind regards,

Richard Mankin, Ph. D.

Academic Editor

PLOS ONE
---

## [Editor Report · Acceptance letter]

5 Nov 2019

PONE-D-19-19263R1 

Antibiotic use in mandarin production (*Citrus reticulata* Blanco) in major mandarin-producing areas in Thailand: a survey assessment 

Dear Dr. Tangcharoensathien:

I am pleased to inform you that your manuscript has been deemed suitable for publication in PLOS ONE. Congratulations! Your manuscript is now with our production department. 

With kind regards,

on behalf of

Dr. Richard Mankin 

Academic Editor

PLOS ONE